

# Unequal contribution of native South African phylogeographic lineages to the invasion of the African clawed frog, *Xenopus laevis*, in Europe

Charlotte De Busschere[1], Julien Courant[2], Anthony Herrel[2], Rui Rebelo[3], Dennis Rödder[4], G. John Measey[5] and Thierry Backeljau[1,6]

[1] Operational Direction Taxonomy and Phylogeny, Royal Belgian Institute of Natural Sciences, Brussels, Belgium
[2] UMR7179, Département d'Ecologie et de Gestion de la Biodiversité, Centre national de la recherche scientifique, Paris, France
[3] Departamento de Biologia Animal/ Centre for Ecology, Evolution and Environmental Changes, Faculdade de Ciências da Universidade de Lisboa, Lisboa, Portugal
[4] Herpetology Department, Zoologisches Forschungsmuseum Alexander Koenig, Bonn, Germany
[5] Centre of Invasive Biology, Department of Botany and Zoology, University of Stellenbosch, Stellenbosch, South-Africa
[6] Evolutionary Ecology Group, University of Antwerp, Antwerp, Belgium

Corresponding author
Charlotte De Busschere,
charlotte.debusschere@gmail.com

## ABSTRACT

Due to both deliberate and accidental introductions, invasive African Clawed Frog (*Xenopus laevis*) populations have become established worldwide. In this study, we investigate the geographic origins of invasive *X. laevis* populations in France and Portugal using the phylogeographic structure of *X. laevis* in its native South African range. In total, 80 individuals from the whole area known to be invaded in France and Portugal were analysed for two mitochondrial and three nuclear genes, allowing a comparison with 185 specimens from the native range. Our results show that native phylogeographic lineages have contributed differently to invasive European *X. laevis* populations. In Portugal, genetic and historical data suggest a single colonization event involving a small number of individuals from the south-western Cape region in South Africa. In contrast, French invasive *X. laevis* encompass two distinct native phylogeographic lineages, i.e., one from the south-western Cape region and one from the northern regions of South Africa. The French *X. laevis* population is the first example of a *X. laevis* invasion involving multiple lineages. Moreover, the lack of population structure based on nuclear DNA suggests a potential role for admixture within the invasive French population.

## INTRODUCTION

Reconstructing the invasion history and dynamics of invasive species is crucial for understanding biological invasions and for developing appropriate management strategies

(*Sakai et al., 2001*; *Lee & Gelembiuk, 2008*; *Prentis et al., 2009*). Moreover, exploring patterns of population genetic variation and evolutionary processes may be key to infer the invasive potential of invasive alien species (*Sakai et al., 2001*; *Lee & Gelembiuk, 2008*). In this study, we focus on the African clawed frog, *Xenopus laevis* (*Daudin, 1802*), which is indigenous to Southern Africa, including South Africa up to Malawi (i.e., *X. laevis* sensu stricto *Furman et al., 2015*). Due to deliberate and accidental introductions from laboratories and pet suppliers, invasive *X. laevis* populations have become established in Asia, Europe, North America and South America (*Tinsley & McCoid, 1996*; *Lobos & Measey, 2002*; *Crayon, 2005*; *Fouquet & Measey, 2006*; *Faraone et al., 2008*; *Measey et al., 2012*; *Peralta-García, Valdez-Villavicencio & Galina-Tessaro, 2014*). Invasive *X. laevis* populations have negative impacts on local biota by reducing the occurrence of reproduction (*Lillo, Faraone & Lo Valvo, 2010*) and increasing predation pressures on native prey organisms (*Lafferty & Page, 1997*; *Faraone et al., 2008*; *Measey et al., 2015*). *Lobos & Measey (2002)* also suggested that *X. laevis* might have indirect impacts on the aquatic system such as increasing water turbidity and nutrient release. Finally, the spread of *Batrachochytrium dendrobatis*, which causes the amphibian skin disease chytridiomycosis and negatively impacts amphibian populations (*Berger et al., 1998*; *Lips et al., 2006*; *Skerratt et al., 2007*; *Voyles et al., 2009*; *Crawford, Lips & Bermingham, 2010*), has been linked to invasive amphibian species such as *X. laevis* which are often asymptotic carriers (*Weldon et al., 2004*); however, this link has yet to be proven. *Measey et al. (2012)* suggested that the global potential invasiveness of *X. laevis* has been severely underestimated and that it is likely that *X. laevis* will expand its present colonized area. For example in Europe, introduced *X. laevis* populations are currently established in France (*Fouquet, 2001*), Portugal (*Rebelo et al., 2010*) and Italy (Sicily) (*Lillo et al., 2005*), though the predicted suitable climate space for *X. laevis* covers over one million km$^2$ making this a species of European concern (*Measey et al., 2012*).

Within its native range, *X. laevis* has a wide geographical distribution in which it occupies a variety of natural, as well as manmade waterbodies (*Measey, 2004*). Native *X. laevis* populations are distributed from winter rainfall regions in the south-western Cape region to summer rainfall regions in the north; and from sea level to nearly 3,000 m in Lesotho (*Measey, 2004*). Furthermore, several physiological and behavioural traits enable *X. laevis* to cope with, dehydration, high levels of salinity, starvation and anoxic conditions (reviewed in *Measey et al., 2012*). Throughout the native range of *X. laevis* significant population differentiation has been observed based on both mitochondrial (mtDNA) and nuclear DNA (nDNA) sequences (*Grohovaz, Harley & Fabian, 1996*; *Measey & Channing, 2003*; *Du Preez et al., 2009*; *Furman et al., 2015*). As such, *Furman et al. (2015)* identified four phylogeographic lineages within South Africa based on mtDNA and nDNA: (1) south-western Cape (SA1–SA2), (2) Beaufort West (SA4), (3) Niewoudtville (SA7) and (4) northern South Africa (Kimberley, Victoria West, Potchefstroom, SA5) (Fig. 1). The latter two clades are geographically separated by the coastal regions due to the Great Escarpment i.e., a plateau edge running parallel with the South African coast and separating the inland plateau from the coastal plains (*Grab, 2010*). An admixture zone was observed around Laingsburg (SA3) between south-western Cape (SA1–SA2) and

Beaufort West (SA4). Laingsburg (SA3) and Beaufort West (SA4) are both located within the lowland part of the Great Karoo (700–800 m above sea level) which is separated from the south-western Cape (SA1–SA2) by the Cape Fold Mountains (*Du Preez et al., 2009*; *Furman et al., 2015*) and from the north by the Great Escarpment (Fig. 1). Another admixture zone was suggested between Niewoudtville (SA7) and the south-western Cape populations (SA1) around Vredendal (*Measey & Channing, 2003*). In its native range *X. laevis* also displays substantial phenotypic population differentiation. *Du Preez et al. (2009)*, for example, showed that male *X. laevis* from south of the Cape Fold Mountains were longer and heavier than males from north of the Cape Fold Mountains.

Given its wide native geographical and ecological ranges and high population genetic diversity, it is important to identify the source areas and population(s) of invasive *X. laevis* populations if one aims to understand the invasion history and invasive potential of this species (*Sakai et al., 2001*; *Lee, 2002*; *Dlugosch & Parker, 2008*; *D'Amen, Zimmermann & Pearman, 2013*). The origin of invasive *X. laevis* populations in Sicily and Chile has already been explored using DNA markers (*Lillo et al., 2013*; *Lobos et al., 2014*). Similarly, the origin of specimens from animal suppliers i.e., *Xenopus*-1 Inc. (Dexter, MI, USA) and *Xenopus* Express (Brooksville, FL, USA), has been assessed (*Du Preez et al., 2009*). These studies support the assumption that export of *X. laevis* specimens for laboratory use mainly stemmed from the south-western Cape region in South Africa (*Tinsley & McCoid, 1996*; *Weldon, De Villiers & Du Preez, 2007*), although other source areas cannot be excluded, especially for older introduced stocks (*Du Preez et al., 2009*; L Van Sittert & J Measey, 2015, unpublished data).

The present paper aims to unravel to which extent single or multiple native phylogeographic lineages have contributed to the invasive populations in France and Portugal by comparing mtDNA and nDNA data sampled across the invaded, as well as the native range. In France, it is assumed that *X. laevis* was introduced at Bouillé-Saint-Paul (Deux Sèvres) from a nearby breeding facility where *X. laevis* was bred from the 1950s until 1996 (*Fouquet, 2001*; *Fouquet & Measey, 2006*). From that breeding facility *X. laevis* may have escaped repeatedly and was probably released when the facility was definitively closed in 1996 (*Measey et al., 2012*). Currently, French *X. laevis* populations occupy an area of approximately 200 km$^2$ near the city of Saumur (Maine-et-Loire) (Fig. 1). The introduction of *X. laevis* in Portugal is assumed to be accidental, caused in 1979 by the inundation of a basement of a research institute at Oeiras along river Laje, about 20 km west of Lisbon (*Rebelo et al., 2010*; *Measey et al., 2012*). Nowadays, several populations are found in two tributaries of river Tagus, i.e., river Laje and river Barcarena (Fig. 1) (*Rebelo et al., 2010*).

## METHODS

### Taxon sampling

In total 80 individuals from 32 localities, covering the known area invaded by *X. laevis* in France (FR) and Portugal (PT) were captured. For comparison with previous work of *Lillo et al. (2013)* we included two specimens from Sicily (IT, provided by Francesco
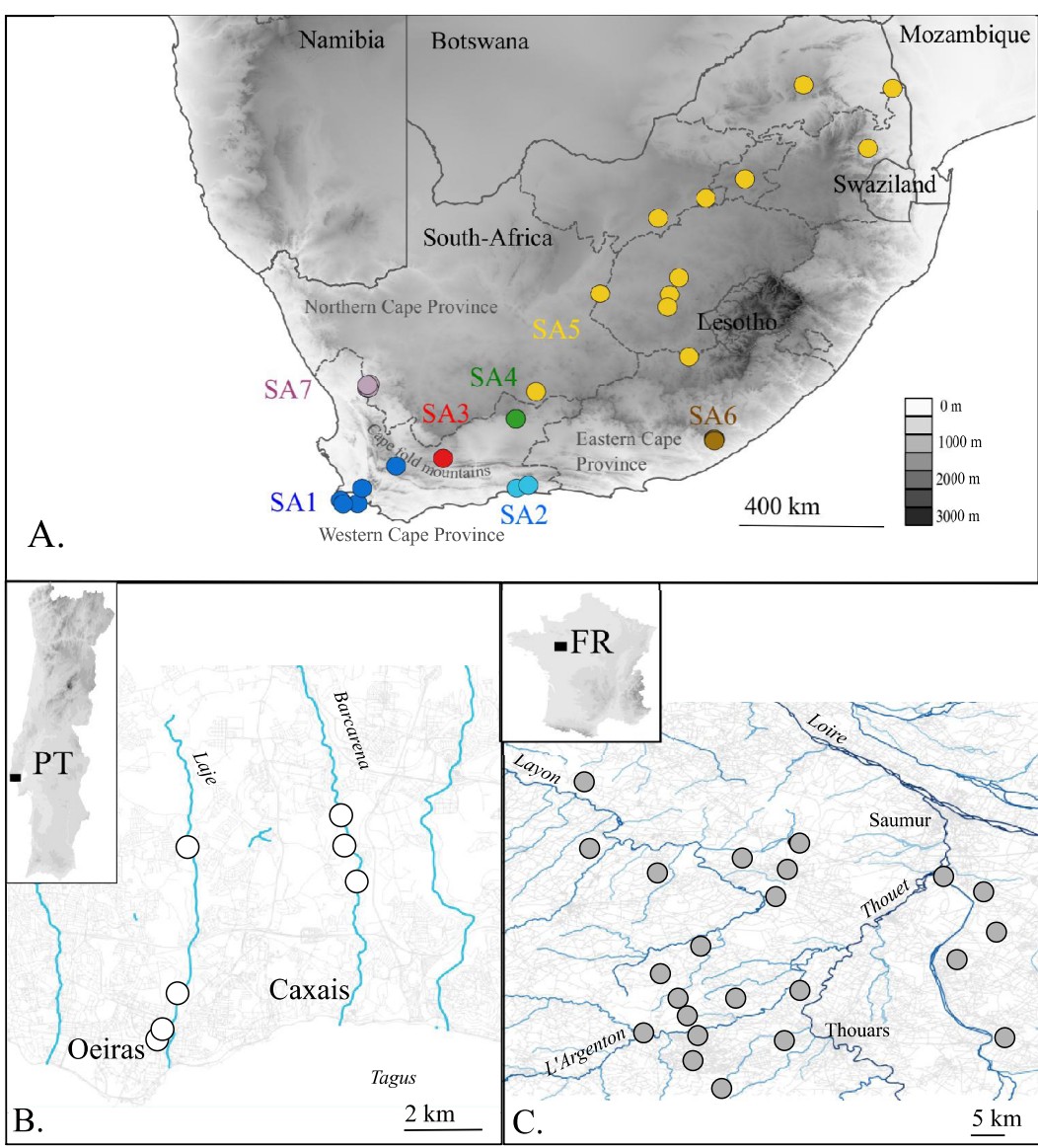

**Figure 1** **Map of the native (A. South Africa) and invaded *X. laevis* localities (B. Portugal, C. France) surveyed in this study.** Abbreviations and colours of sampling localities (circles) refer to geographical regions that are mentioned in methods (see 'Taxon sampling'). More detailed locality information is provided in Online Resource 1. National and provincial borders of South African Provinces are visualized by solid and dashed lines respectively (A). Rivers and roads are represented by blue and grey lines respectively (B) and (C). Names of main rivers (italic) and towns are shown (B) and (C).

Lillo). Within the native range of *X. laevis*, 21 specimens were sampled. Seven specimens came from two localities in the south-western Cape (SA1) i.e., Cape of Good Hope nature reserve and a single dam at Jonkershoek, the location from which animals were shipped for international trade from the 1940s to 1970s (L Van Sittert & J Measey, 2015, unpublished data). 14 specimens came from five localities around Rooikrantz Dam near the historical Pirie hatchery nearby King Williams Town, Eastern Cape (SA6) (L Van Sittert

& J Measey, 2015, unpublished data). These new data were supplemented by data on 164 South African specimens available on GenBank. One individual of *Xenopus gilli Rose & Hewitt, 1927* was sampled near Kleinmond (South Africa). Based on *Furman et al. (2015)*, sampling localities in South Africa were grouped into seven geographical regions: SA1: south-western Cape, south-west of the Cape Fold Mountains up to De Doorns; SA2: Cape, Hoekwil & Tsitsikamma region; SA3: Cape, Laingsburg; SA4: Cape, Beaufort West; SA5: northern South Africa (Kimberley, Victoria West, Potchefstroom); SA6: Cape, Rooikrantz Dam; SA7: Nieuwoudtville (Fig. 1). All specimen data with full locality information are provided in Online Resource 1. Animals from the Portuguese invasive population were captured under the permit no 570/2014/CAPT from Instituto da Conservação da Natureza e das Florestas, in the scope of the "Plano de erradicação de *Xenopus laevis* nas ribeiras do Concelho de Oeiras". Animals from the native South-African populations were sampled under the permits 0056-AAA007-00092 (CapeNature, SA1) and CRO 109/13CR (Eastern Cape, SA6) provided by the Department of Economic Development, Environmental Affairs and Tourism and with ethics approval from the Research Ethics Committee: Animal Care and Use (protocol number: SU-ACUD14-00028).

## DNA amplification and sequencing

Invasive specimens were euthanized with a lethal injection of sodium pentobarbital. Muscle tissue was dissected from invasive specimens, while native wild caught specimens were toe-clipped to obtain tissue samples. Genomic DNA was extracted by means of a NucleoSpin® tissue kit (Macherey-Nagel, Düren, Germany) according to the manufacturer's protocol. Five genomic DNA fragments (circa (ca.) 2,040 bp) representing two mitochondrial and three nuclear gene fragments were amplified using PCR. These genes were selected in order to enable comparison with previously published work on native (*Measey & Channing, 2003*; *Du Preez et al., 2009*; *Bewick, Anderson & Evans, 2011*; *Furman et al., 2015*) and invasive *X. laevis* specimens (*Lillo et al., 2013*; *Lobos et al., 2014*). Fragments of the mitochondrial cytochrome b gene (Cytb; ca. 280 bp) and 16S ribosomal DNA (16S; ca. 800 bp) were amplified and sequenced with the primer pairs Cytb I /Cytb II and 16Sc-L/16Sd-H (*Kessing et al., 1989*; *Evans et al., 2003*). Fragments of the nuclear protein coding genes arginine methyltransferase 6 (Prmt6; ca. 666 bp), androgen receptor isoform α (AR; ca. 402 bp) and microtubule associated serine/threonine kinase-like protein (Mastl; ca. 539 bp) were amplified and sequenced with the primer pairs Exon4_for1/Exon4_rev2, XLAR_for_40/XLAR_rev_431 and Exon13_fora/Exon13_reva (*Bewick, Anderson & Evans, 2011*). The nuclear primer pairs are assumed to be paralog-specific, hence only amplifying one pair of alleles (see *Bewick, Anderson & Evans, 2011*). PCR amplifications were run with the conditions reported in Online Resource 3. PCR products were purified with FastAP TM thermosensitive alkaline phosphatase in combination with exonuclease I and subsequently sequenced in both directions. Nucleotide sequences were assembled and edited in CodonCode Aligner (CodonCode Corporation, Dedham, MA, USA). They were aligned together with corresponding well-documented sequences of South African *X. laevis* available from GenBank using the ClustalW algorithm (*Thompson, Higgins & Gibson, 1994*) in *MEGA* v. 6 (*Tamura et al., 2013*). For comparison, the two Cytb haplotypes found by

*Lobos et al. (2014)* in Chilean (CL) invasive *X. laevis* populations were included in the Cytb alignment. Cytb and 16S gene fragments obtained by *Lillo et al. (2013)* were not included in the current alignments as they only partially overlapped due to the use of different primers. Nuclear sequences were converted into haplotypes using the PHASE algorithm (*Stephens, Smith & Donnelly, 2001*; *Stephens & Donnelly, 2003*) implemented in DnaSP v. 5 (*Librado & Rozas, 2009*). All new sequence data were deposited in Genbank (accession numbers in Online Resource 1: https://figshare.com/s/44b71de473fc11e5829d06ec4b8d1f61).

## Genetic diversity

The following indices of genetic diversity were estimated using DnaSP v. 5 (*Librado & Rozas, 2009*): numbers of different haplotypes ($h$), the number of segregating sites ($S$), nucleotide diversity ($\pi$, i.e., the average number of nucleotide differences per site between two sequences) and haplotype/allelic diversity ($Hd$, i.e., the probability that two haplotypes drawn uniformly at random from a population are not the same).

## mtDNA analysis

Phylogenetic relationships among invasive and native *X. laevis* mtDNA sequences were reconstructed with Bayesian inference (BI) and Maximum Parsimony (MP). Three alignments were created: two separate Cytb and 16S alignments including only unique alleles and a concatenated Cytb-16S alignment including only unique haplotypes. In all analyses, *X. gilli* was used as outgroup. Parsimony informative sites were calculated with DnaSP v. 5 (*Librado & Rozas, 2009*). BI analyses were performed using MrBayes v. 3.2.4 (*Ronquist & Huelsenbeck, 2003*). Cytb, 16S and Cytb-16S alignments were analysed under a general time reversible (GTR) model with all model parameters estimated from the data and a proportion of invariant sites (+I) as selected by jModeltest v. 2.1.7 (*Posada, 2008*). BI analyses were run with two different Metropolis-coupled Markov chains for 10 million generations with sampling every 1,000th generation. The average standard deviations of split frequencies, the potential scale reduction factors and the plots of likelihood versus generation were evaluated to ensure convergence based on the log file and using Tracer v1.6 (*Rambaut et al., 2013*). A total of 25% of the trees were discarded as burn-in and posterior probabilities were calculated for each split from the remaining set of trees. MP trees were estimated with a neighbour joining tree as starting tree using the Phangorn package (*Schliep, 2011*) in R 3.1.1 (*R Development Core Team, 2014*). A set of most-parsimonious trees was generated using the parsimony ratchet (*Nixon, 1999*) with nearest neighbour interchange rearrangement, 10,000 ratchet iterations and up to maximum 10 rounds. Parsimony bootstrap values were obtained with 1,000 bootstrap replicates using the bootstrap.phyDat function in R 3.1.1. All topologies were visualized and edited in respectively Figtree v. 1.4.2 (*Rambaut, 2014*) and TreeGraph2 v. 2.4.0-456 beta (*Stöver & Müller, 2010*). Differentiation between native geographical areas and invaded regions was quantified using pairwise *Fst* values computed based upon nucleotide pairwise distances and tested for significance with 999 permutations using Arlequin v. 3.5.3.1 (*Excoffier & Lischer, 2010*). The population comparison was done solely for the 16S alignment as this dataset included sequences of most individuals ($n = 179$) and representatives of all populations.

## Autosomal analysis

The minimum number of recombination events within alleles were estimated in DnaSP v.5 (*Hudson & Kaplan, 1985*; *Librado & Rozas, 2009*). Given putative recombination, median-joining networks (MJN) (*Bandelt, Forster & Rohl, 1999*) were constructed in PopART v.1.7.2 (PopART, 2015) for each nuclear locus separately to illustrate the mutational differences between alleles and their distribution among invasive and native populations (*Mardulyn, 2012*). Pairwise *Fst* values were computed based on the allele frequencies to show to what extent alleles are differently sorted among populations (e.g., where the same alleles are found in different populations but with different frequencies). Native populations were defined based upon the geographical regions mentioned in 'Taxon sampling.' Individuals from France and Portugal respectively were treated as two populations. Pairwise *Fst* values were computed from the allele frequencies with 999 permutations to generate the probability that a random value would be greater than or equal to the observed data using GenAlEx v. 6.5 (*Peakall & Smouse, 2012)*. Individuals with missing data for more than one gene were excluded from the analysis. In order to visualize population differentiation, the pairwise *Fst* matrix was subsequently used as a distance matrix for a principal coordinates analysis (PCoA) in GenAlEx v.6.5.

## RESULTS

### mtDNA

Mitochondrial alignments were constructed for Cytb and 16S involving 101 and 181 individuals respectively. The concatenated Cytb-16S alignment involved 64 individuals. None of the alignments showed gaps. The Cytb alignment involved16 unique alleles of *X. laevis* with 42 variable positions of which 34 were parsimony informative. No Cytb sequences were obtained of specimens from the regions SA2, SA3 and SA4. In contrast, all geographic regions were represented in the 16S alignment, which involved 15 alleles of *X. laevis* with 25 variable positions of which 17 were parsimony informative. The concatenated Cytb-16S alignment involved seven unique *X. laevis* haplotypes with 43 variable positions of which 39 were parsimony informative (Online Resource 2: https://figshare.com/s/44b71de473fc11e5829d06ec4b8d1f61). The concatenated alignment comprised only specimens from the invaded regions PT, IT and FR and the native regions SA1 and SA6. The BI and MP trees for Cytb and 16S are shown in Fig. 2. These analyses all resolved consistently three well-supported clades comprising *X. laevis* alleles from the geographic regions (1) SA5, SA6 and FR, (2) SA7 and (3) SA1, PT, FR, IT. Additionally, the phylogeny generated from the 16S data strongly supported (1) SA1, SA2, SA3, FR, PT and IT and (2) SA3-SA4 as well-supported *X. laevis* clades. BI and MP trees for the concatenated data (Online Resource 2) showed two highly supported clades comprising haplotypes from (1) FR, PT, IT, SA1 and (2) FR and SA6. Pairwise population *Fst* values among all geographic regions based on 16S were significant except among (1) SA1, SA2 and PT, (2) SA5 and SA6 and among (3) FR and SA6 (Online Resource 4).

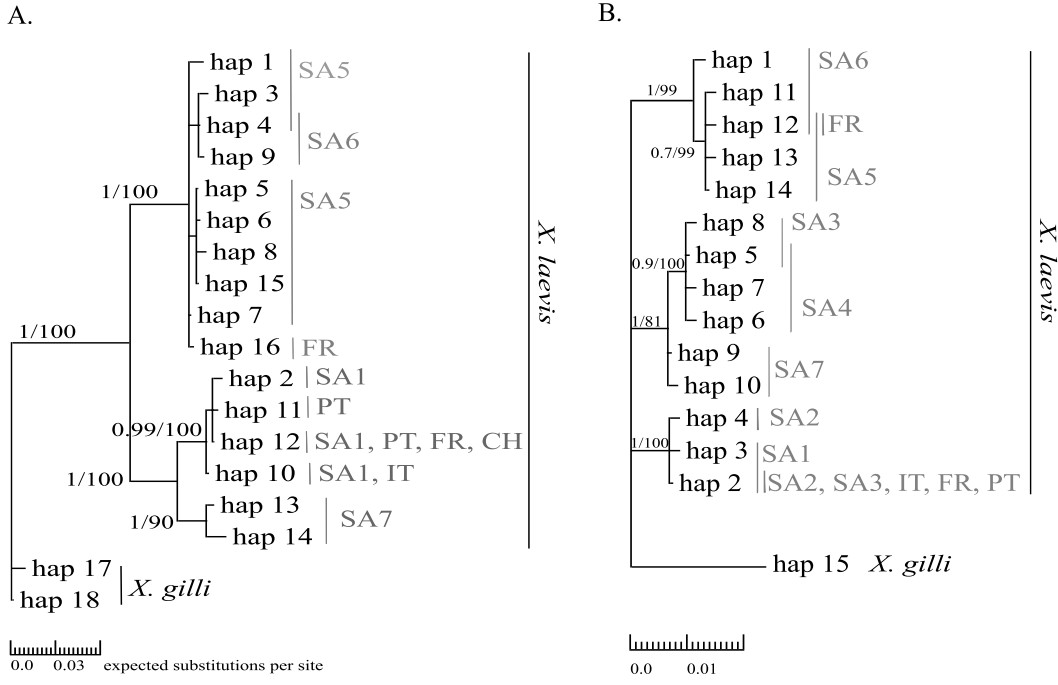

**Figure 2** **MP and BI inference based on Cytb (A) and 16S alignments (B).** Bayesian consensus trees are visualized with Posterior BI bootstrap ($B > 0.70$) and Parsimony bootstrap values. Parsimony scores (i.e., tree length) of Cytb MP tree and 16S MP tree were 83 and 47 respectively. Geographical regions where alleles have been observed are indicated in grey (abbreviations see Fig. 1 and 'Methods').

## nDNA

AR sequences ($n = 121$) showed eight polymorphic positions of which seven revealed heterozygous genotypes. The maximum number of heterozygous positions observed within a single AR sequence was three but that was only observed in two individuals. The two most frequent AR alleles i.e., hap_1 and hap_2 differed by two nonsynonymous substitutions. The Mastl alignment ($n = 180$) showed 37 polymorphic positions and 108 individuals were heterozygous at $3 \pm 2$ positions on average. The Prmt6 alignment ($n = 176$) showed 20 polymorphic positions and 107 individuals were heterozygous at on average $2 \pm 1$ positions. MJN revealed high allelic diversity at Mast1 and Prmt6 and low allelic diversity at AR (Table 1). In the MJN, the most frequent alleles were often found in different geographical regions (Fig. 4). When combining the allelic frequency information of the three nDNA genes, pairwise population *Fst* values among all geographic regions were significant except between SA5 and SA6 (Online Resource 5: https://figshare.com/s/44b71de473fc11e5829d06ec4b8d1f61). The first two PCoA-axes explained 75% of the total variance from the pairwise *Fst* values. There was a clear separation among SA7 and the remnant populations along the first PCoA axis (explaining 28% of the total variance). Negative values along the second PCoA axis (explaining 47% of the total variance) were linked to the northernmost up to the north-eastern native populations (SA4–SA7), while positive values were linked to the invasive (Fr, PT) and southernmost
**Table 1 Summary of diversity statistics for all loci and each population.** Statistics describing the number of nucleotides sequenced ($nt$), number of individuals (ind), number of different haplotypes ($h$), number of segregating sites ($S$), nucleotide diversity ($\pi$) and haplotype diversity ($Hd$).

| Gene | Country | Pop | $nt$ | ind | $h$ | $S$ | $\pi$ | $Hd$ |
|------|---------|-----|------|-----|-----|-----|-------|------|
| Cytb | France | FR | 282 | 42 | 2 | 26 | 0.0233 | 0.251 |
| | Portugal | PT | 282 | 18 | 2 | 1 | 0.00105 | 0.294 |
| | South Africa | SA1 | 282 | 12 | 3 | 3 | 0.00265 | 0.53 |
| | | SA5 | 282 | 13 | 8 | 10 | 0.01099 | 0.91 |
| | | SA6 | 282 | 6 | 2 | 1 | 0.00119 | 0.333 |
| | | SA7 | 282 | 4 | 2 | 6 | 0.01429 | 0.667 |
| 16S | France | FR | 544 | 56 | 2 | 13 | 0.00598 | 0.249 |
| | Portugal | PT | 544 | 16 | 1 | 0 | 0 | 0 |
| | South Africa | SA1 | 544 | 23 | 2 | 1 | 0.00055 | 0.3 |
| | | SA2 | 544 | 14 | 2 | 1 | 0.00026 | 0.143 |
| | | SA3 | 544 | 9 | 3 | 11 | 0.00756 | 0.556 |
| | | SA4 | 544 | 12 | 3 | 2 | 0.00106 | 0.53 |
| | | SA5 | 544 | 13 | 3 | 2 | 0.0008 | 0.41 |
| | | SA6 | 544 | 14 | 3 | 3 | 0.00079 | 0.275 |
| | | SA7 | 544 | 22 | 2 | 1 | 0.00068 | 0.368 |
| AR | France | FR | 294 | 58 | 2 | 2 | 0.00338 | 0.495 |
| | Portugal | PT | 294 | 18 | 1 | 0 | 0 | 0 |
| | South Africa | SA1 | 294 | 27 | 1 | 0 | 0 | 0 |
| | | SA2 | 294 | 13 | 4 | 4 | 0.0044 | 0.582 |
| | | SA3 | 294 | 18 | 2 | 2 | 0.00347 | 0.508 |
| | | SA4 | 294 | 20 | 3 | 2 | 0.0014 | 0.229 |
| | | SA5 | 294 | 25 | 3 | 2 | 0.00078 | 0.222 |
| | | SA6 | 294 | 9 | 1 | 0 | 0 | 0 |
| | | SA7 | 294 | 23 | 2 | 1 | 0.00373 | 0.043 |
| Mastl | France | FR | 525 | 53 | 13 | 6 | 0.00499 | 0.804 |
| | Portugal | PT | 525 | 18 | 2 | 5 | 0.00053 | 0.056 |
| | South Africa | SA1 | 525 | 27 | 10 | 10 | 0.00609 | 0.73 |
| | | SA2 | 525 | 12 | 9 | 8 | 0.00662 | 0.873 |
| | | SA3 | 525 | 9 | 8 | 12 | 0.00712 | 0.895 |
| | | SA4 | 525 | 13 | 8 | 9 | 0.00453 | 0.745 |
| | | SA5 | 525 | 13 | 13 | 10 | 0.00452 | 0.938 |
| | | SA6 | 525 | 9 | 13 | 6 | 0.00421 | 0.961 |
| | | SA7 | 525 | 24 | 7 | 10 | 0.00264 | 0.543 |
| Prmt6 | France | FR | 396 | 51 | 16 | 8 | 0.00679 | 0.839 |
| | Portugal | PT | 396 | 18 | 4 | 3 | 0.00213 | 0.605 |
| | South Africa | SA1 | 396 | 27 | 10 | 7 | 0.00375 | 0.829 |
| | | SA2 | 396 | 13 | 9 | 6 | 0.00447 | 0.871 |
| | | SA3 | 396 | 8 | 10 | 9 | 0.00787 | 0.942 |
| | | SA4 | 396 | 12 | 7 | 7 | 0.00498 | 0.79 |
| | | SA5 | 396 | 13 | 13 | 8 | 0.00679 | 0.911 |
| | | SA6 | 396 | 10 | 15 | 13 | 0.00793 | 0.974 |
| | | SA7 | 396 | 22 | 1 | 0 | 0 | 0 |
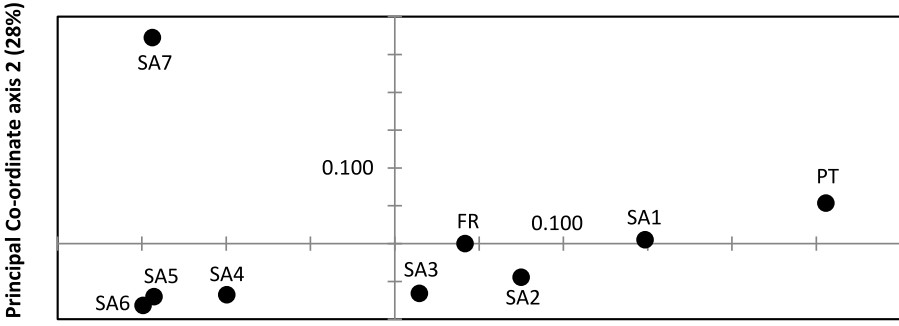

**Figure 3** Result of Principal Co-ordinate analysis of nuclear genetic variation among native (SA1–SA7) and invasive French (FR) and Portuguese (PT) *X. laevis* populations. PCoA of pairwise *Fst* values based on allele frequencies in three nuclear loci ($n = 180$ individuals; Online Resource 5). Abbreviations refer to geographical regions (see Fig. 1).

native populations (SA1–SA3) (Fig. 3). Native populations SA5 and SA6 clustered together (Fig. 3).

## South Africa

Haplotype diversity of nDNA and mtDNA within the native range of *X. laevis* was on average $0.60 \pm 0.20$ and $0.45 \pm 0.18$ respectively. Mitochondrial nucleotide diversity within the native regions was either rather low ($\pi \leq 0.0016$: SA1, SA2, SA4, SA6) or high ($\pi \geq 0.0059$: SA3, SA5, SA7). As mentioned previously, mtDNA and nDNA variation was geographically structured within South Africa (Fig. 2, Online Resource 4 and 5) (*Furman et al., 2015*). The monophyly of the sequences from northern South African regions (SA5) and the newly sampled sites at Rooikrantz Dam in the Eastern Cape (SA6) was consistently strongly supported by both Cytb and 16S (Fig. 2). Furthermore, the pairwise population *Fst* values based on nDNA and on 16S separately were not significant between these two regions (Online Resource 4 and 5: https://figshare.com/s/44b71de473fc11e5829d06ec4b8d1f61).

## Portugal

Nucleotide and haplotype diversities in Portugal were extremely low for all markers ($0 \leq \pi < 0.00105$, $0 \leq Hd < 0.294$, Table 1), except for Prmt6 ($\pi = 0.00213$, Hd = 0.605, Table 1). Only two alleles were exclusively found in Portugal, viz. one for Cytb (hap_11) and one for Prmt6 (hap_27). The concatenated Cytb-16S haplotype (hap_11_2) was only found in Portugal. All other mtDNA and nDNA alleles occurred also within native South African populations. More precisely, identical mtDNA was found in *X. laevis* populations from the south-western Cape SA1 (Cytb: hap_12) and from the Cape regions SA1–SA3 (16S: hap_2). Identical AR, Prmt6 and Mastl alleles were found in respectively the following native regions (1) SA1–SA3, (2) SA1–SA4 and SA7 and (3) SA1–SA7. Pairwise *Fst* values based on 16S were not significant among Portugal and native regions SA1 and SA2 (Online Resource 4). PCoA and MJN revealed that nuclear allele frequencies within Portugal were most similar to those of native region SA1 (Fig. 3, Online Resource 5). The mean Portuguese nucleotide diversity ($\pi = 0.0007 \pm 0.0009$) across all loci was much lower than

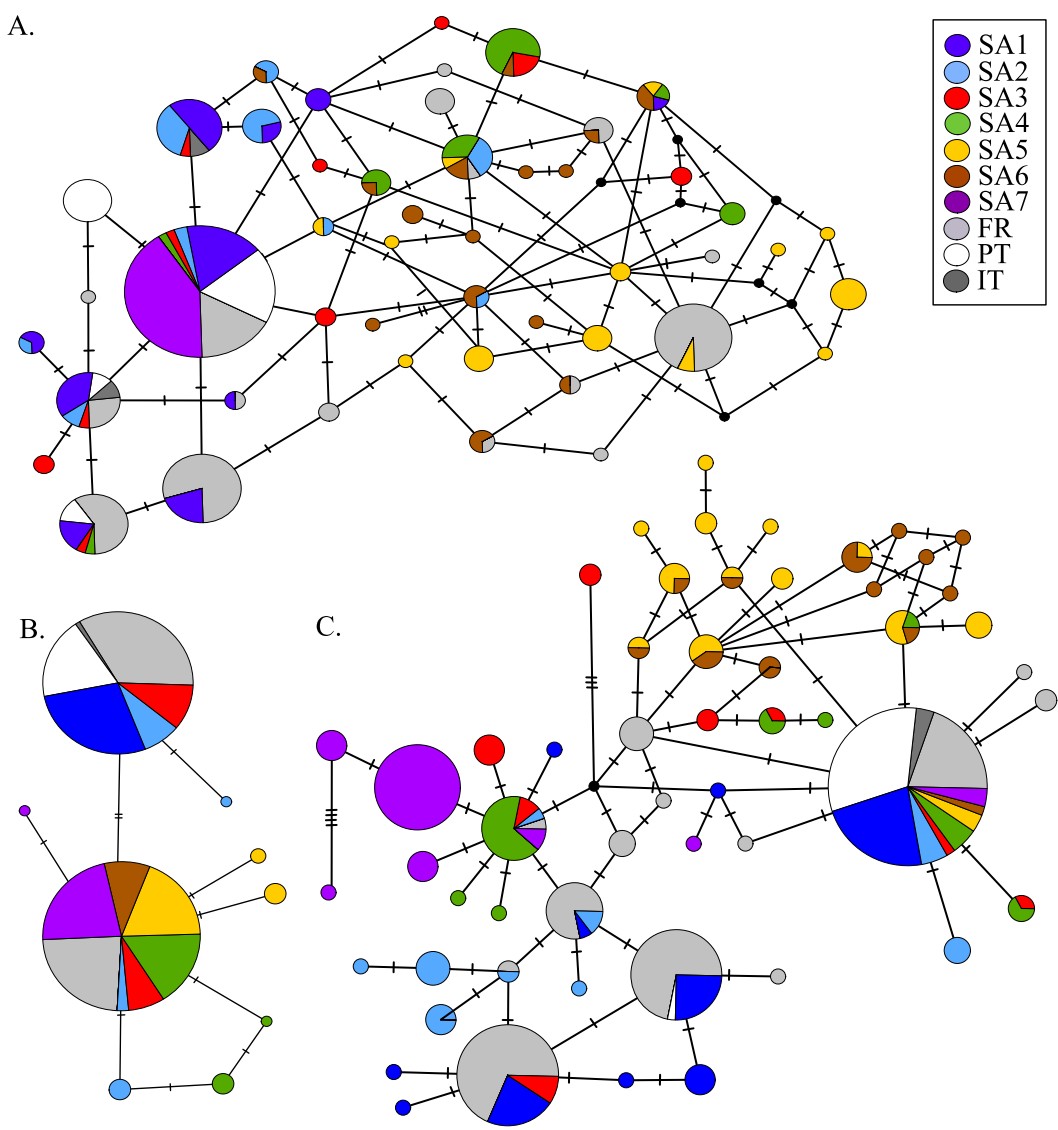

**Figure 4   MJN of nuclear Prmt6 (A), AR (B) and Mastl (C) sequence data from native and invasive *X. laevis* populations.** The sizes of the circles are proportional to allele frequencies. Colours refer to native geographic and invaded regions (see legend). Small black nodes represent unsampled alleles and numbers of mutations are marked by stripes on the connecting branches.

in native populations ($\pi = 0.0039 \pm 0.0035$). Similarly, the mean Portuguese haplotype diversity ($Hd = 0.191 \pm 0.261$) across all loci was lower than in the native populations ($Hd = 0.558 \pm 0.323$).

## France

Two distinct alleles were observed in the introduced French population for both Cytb and 16S (Table 1, Fig. 2). Moreover, these alleles were highly similar or identical to alleles found in two geographically non-overlapping native regions in South Africa, i.e., the northern South Africa and Rooikrantz Dam region (SA5–SA6) and the south-western Cape region (SA1–SA3) (Fig. 2). The most abundant Cytb allele in France (i.e., hap_16;

86% of specimens) was very similar to Cytb sequences in northern South Africa and Rooikrantz Dam populations (SA5–SA6) with on average four nucleotide differences (Fig. 2). Similarly, 86% of the French individuals had a 16S allele (i.e., hap_12) identical to a 16S sequence only found in northern South Africa and Rooikrantz Dam region (SA5–SA6)(Fig. 2). The concatenated Cytb-16S dataset comprised 38 French individuals which represented two different haplotypes. One haplotype was identical to a haplotype found in SA1 (i.e., hap_12_2) and the other haplotype was only found in France but highly similar to haplotypes found in SA6 (i.e., hap_16_12; Online Resource 2).

MtDNA haplotype diversity for each marker was low within the French population ($Hd \leq 0.251$, Table 1). Pairwise $Fst$ value based on 16S among SA6 and France was not significant ($Fst = 0.049$; Online Resource 4). Haplotype diversities for nDNA ranged from 0.50 for AR up to 0.80 for Mastl and Prmt6 (Table 1). France comprised 13 Mastl and 16 Prmt6 alleles with seven Mastl and 10 Prmt6 alleles being hitherto only found in France. nDNA allele frequencies in France were most similar to those in native populations from Laingsburg in the south-western Western Cape Province (SA3, Fig. 3, Online Resource 4: https://figshare.com/s/44b71de473fc11e5829d06ec4b8d1f61). No significant difference was observed when comparing nDNA allele frequencies among individuals representing the two different mitochondrial groups ($Fst = 0$; $p = 0.421$), however sample sizes were low i.e., 49 and 8 individuals representing northern South Africa-Rooikrantz Dam mtDNA (SA5–SA6) and south-western Cape mtDNA (SA1–SA2) respectively. In comparison with the native geographic regions, mean nucleotide diversity across all loci in France was higher ($\pi = 0.009 \pm 0.008$ versus $\pi = 0.0039 \pm 0.0035$) while French haplotype diversity across all loci was comparable ($Hd = 0.528 \pm 0.287$ versus $Hd = 0.558 \pm 0.323$).

### Sicily

The Sicilian individuals were identical to native individuals from SA1 for Cytb and from SA1–SA3 for 16S (Fig. 1). Only one concatenated Cytb-16S haplotype was found which was unique to Sicily and highly similar to haplotypes found in FR, PT and SA1. Sicilian AR and Prmt6 alleles were identical to alleles occurring in the native range regions SA1-SA3. Cytb and 16S sequences of Sicilian individuals sampled by *Lillo et al. (2013)* were not included in the current analyses as they involved different gene regions. However, the mtDNA sequences in the present study were identical to those of *Lillo et al. (2013)* in the overlapping gene regions (Cytb: ~242 bp, 16S: ~374 bp).

## DISCUSSION

Since the 1930s, *X. laevis* has successfully invaded extensive areas worldwide particularly due to its popularity for laboratory use and pet trade (*Tinsley, Loumont & Kobel, 1996*; *Gurdon & Hopwood, 2000*; *Lobos & Measey, 2002*; *Crayon, 2005*; *Faraone et al., 2008*; *Measey et al., 2012*; *Herrel & Van Der Meijden, 2014*). Reconstructing the invasion history of *X. laevis* is pivotal to understand the invasion biology and range dynamics of this species and as such, might be critical for developing future management strategies (*Sakai et al., 2001*; *Lee & Gelembiuk, 2008*; *Prentis et al., 2009*). The identification of source populations is particularly relevant when there is extensive population differentiation within the

native range because, as well phenotypic as genotypic traits of colonizing individuals might influence the invasion process (*Sakai et al., 2001*; *Lee & Gelembiuk, 2008*). Here, the origin of invasive French and Portuguese *X. laevis* specimens was investigated using DNA sequence data.

The genetic diversity of the invasive Portuguese populations was lower than across the native range of *X. laevis* in South Africa, but the Portuguese sequences were very similar or even identical to those of native individuals from the south-western Cape region in South Africa. This suggest (1) that the Portuguese populations may be derived from the latter and (2) that the Portuguese population derives from a single colonization event involving a small number of individuals, most likely stemming from one and the same source population in the south-western Cape region. This is in line with what one would expect from the historical data which attributes the introduction of *X. laevis* in Oeiras (Portugal) to a single accidental flood of a basement of a research institute in 1979 (*Rebelo et al., 2010*). Analogous to the Portuguese samples, the identical sequences shared by the Sicilian (*Lillo et al., 2013*), Chilean (*Lobos et al., 2014*) and south-western Cape populations samples support the idea that *X. laevis* was imported in Sicily and Chile from wild populations in the south-western Western Cape Province. Indeed, the export of *X. laevis* from the Western Cape Province is well documented, especially between 1940 and 1974 when there was only one official supplier in South Africa, i.e., Jonkershoek Fish Hatchery (*Weldon, De Villiers & Du Preez, 2007*).

In contrast to the *X. laevis* colonization in Portugal, two distinct and divergent mtDNA lineages were detected in France. These lineages were related to two geographically non-overlapping native regions in South Africa. A majority of the French individuals possessed mtDNA highly similar or identical to a phylogeographic lineage from northern South Africa and Rooikrantz Dam (SA5–SA6, further referred to as the northern lineage). The other French individuals had mtDNA identical to native individuals from the south-western Cape (SA1–SA2). Haplotype and sequence diversity in the French population were relatively high and comparable to those in native South African regions. The mtDNA data thus suggest that two distinct phylogeographic lineages i.e., south-western Cape and northern lineage, contributed to the invasion of *X. laevis* in France. Although the nDNA data are consistent with this suggestion, they solely indicate the Laingsburg region (SA3) and the south-western Cape (SA1–SA2) as possible source areas of the French nDNA alleles. This might be explained by either a scenario of a single South African source area where south-western and northern lineages admix or a scenario in which France was invaded by individuals from several distinct South African source areas followed by admixture of the colonizing animals. In the case of the first scenario, the most likely source population based upon the nDNA data would be the Laingsburg region which is considered as an admixture zone among the south-western Cape regions SA1-SA2 and the Beaufort West region SA4 (*Du Preez et al., 2009*; *Furman et al., 2015*). However, individuals from Laingsburg do not show mtDNA of the northern populations. Hence, the occurrence of the main mitochondrial alleles (Cytb: hap_16; 16S: hap_12) in French individuals cannot be explained by this scenario. Conversely, historical data supports the second scenario in which the *X. laevis* invasion in France stems from different source areas as animal suppliers are known from

both the south-western Western Cape province and the Eastern Cape Province (Port Elizabeth)(L Van Sittert & J Measey, 2015, unpublished data). Unfortunately, information concerning export from the latter region is limited. In contrast, there is some export data from the Western Cape province in the period that the French breeding facility was active (1950–1996) (*Weldon, De Villiers & Du Preez, 2007*). Until 1974, there was one official animal supplier i.e., Jonkershoek Fish Hatchery. Subsequently, trading was left to private enterprises for which export and collection information is hitherto unknown. Yet, since 1990, permits were licenced to four South African animal suppliers restricting the collection of *X. laevis* to man-made water bodies (*Weldon, De Villiers & Du Preez, 2007*). Around the same time, in 1989, the ownership of the French breeding facility changed (*Measey et al., 2012*). Taking all these circumstances together, it seems likely that during its 56 years of existence, the French breeding facility might have imported *X. laevis* repeatedly from different South African sources. Moreover, it seems not unlikely that secondary trading occurred within Europe or even worldwide e.g., among commercial breeding facilities and/or research institutes.

To the best of our knowledge, the French *X. laevis* population is the first example in which two geographically non-overlapping phylogenetic lineages participated in a *X. laevis* invasion. As mentioned previously, two distinct mtDNA lineages, related to on the one hand the northern regions of South Africa and on the other hand the southwestern Cape regions, are present in the French population. However, no population structure could be found in the invasive French populations based on nDNA. In contrast, within the native South African range these mtDNA lineages are significantly divergent based on nDNA. The latter observation suggests a potential role for ongoing admixture within the invasive French population. Likewise, admixture might have occurred within breeding facilities among specimens from different source populations preceding the French invasion. The combination of genetic variation from multiple phylogeographic lineages might explain the high level of genetic diversity within France, which was comparable to, or even higher than, the diversity observed in native regions. An increase in genetic diversity within invasive population relative to native populations due to multiple introductions of genetically divergent source populations has also been demonstrated in invasions of other organisms elsewhere (*Novak & Mack, 1993*; *Kolbe et al., 2004*; *Kolbe et al., 2008*; *Lavergne & Molofsky, 2007*). Moreover, intraspecific admixture among previously isolated multiple divergent genetic lineages is often suggested to play an important role in driving the success of colonising populations (*Sakai et al., 2001*; *Lavergne & Molofsky, 2007*; *Kolbe et al., 2008*; *Lee & Gelembiuk, 2008*; *Rius & Darling, 2014*). Yet, a species' invasiveness is not only function of the amount of genetic variation, but even more importantly of the nature of adaptive genetic variation (*Dlugosch et al., 2015*). Concerning *X. laevis*, it is clear that single introduction events even with low levels of genetic diversity have proven to be highly successful in invading non-native areas such as Chile and Sicily. However, in order to predict the species' potential range expansion the intraspecific variation should be taking into account (*Pearman et al., 2010*; *D'Amen, Zimmermann & Pearman, 2013*), especially for the French population, in which two genetically distinct and environmentally divergent South African phylogeographic lineages have contributed to the invasion. Hence,

the French population offers a study system for investigating the extent to which the combination of genetic variation from divergent phylogeographic lineages might have influenced the French invasion and/or might influence its future range expansion. In sum, the genetic structure of *X. laevis* in its native South African range allowed us to investigate the geographic origins of invasive *X. laevis* populations in France and Portugal. The current analyses showed that native phylogeographic lineages are not equally represented in invasive European *X. laevis* populations.

## ACKNOWLEDGEMENTS

This work was conducted in the contexts of the Biodiversa project BR/132/A1/INVAXEN-BE "Invasive biology of *Xenopus laevis* in Europe: ecology, impact and predictive models" and of the BELSPO-IUAP project P7/04 "SPEEDY: SPatial and environmental determinants of Eco-Evolutionary DYnamics: anthropogenic environments as a model" and FWO research community W0.009.11N "Belgian Network for DNA Barcoding," coordinated by the "Joint Experimental Molecular Unit (JEMU)" at RBINS. JM would like to thank the DST-NRF Centre of Excellence for Invasion Biology for support. We gratefully thank Francesco Lillo, François Lefebvre, Jean Secondi, Mohlamatsane Mokhatla and Lubabalo Mofu for their aid in sampling Xenopus specimens.

### Funding

This work was financially supported by the Biodiversa project BR/132/A1/INVAXEN-BE "Invasive biology of Xenopus laevis in Europe: ecology, impact and predictive models" at RBINS. JM received funds from the National Research Foundation (South Africa) CPRR13080726510 and incentive funding. The funders had no role in study design, data collection and analysis, decision to publish, or preparation of the manuscript.

### Grant Disclosures

The following grant information was disclosed by the authors:
Biodiversa project: BR/132/A1/INVAXEN-BE.
National Research Foundation (South Africa): CPRR13080726510.

### Competing Interests

John Measey is an Academic Editor for PeerJ.

### Author Contributions

- Charlotte De Busschere conceived and designed the experiments, performed the experiments, analyzed the data, wrote the paper, prepared figures and/or tables.
- Julien Courant, Anthony Herrel, Rui Rebelo and G. John Measey contributed reagents/materials/analysis tools, reviewed drafts of the paper.
- Dennis Rödder reviewed drafts of the paper.
- Thierry Backeljau conceived and designed the experiments, contributed reagents/materials/analysis tools, wrote the paper, reviewed drafts of the paper.

## Animal Ethics

The following information was supplied relating to ethical approvals (i.e., approving body and any reference numbers):

Animals from the Portuguese invasive population were captured under the permit no 570/2014/CAPT from Instituto da Conservação da Natureza e das Florestas, in the scope of the "Plano de erradicação de Xenopus laevis nas ribeiras do Concelho de Oeiras." Animals from the native South-African populations were sampled under the permits 0056-AAA007-00092 (CapeNature, SA1) and CRO 109/13CR (Eastern Cape, SA6) provided by the Department of Economic Development, Environmental Affairs and Tourism and with ethics approval from the Research Ethics Committee: Animal Care and Use (protocol number: SU-ACUD14-00028).

## DNA Deposition

The following information was supplied regarding the deposition of DNA sequences:

New sequences are available from Genbank with accession numbers from KT586615 to KT587048.

## Data Availability

Data and online resource can be found at:

https://figshare.com/s/44b71de473fc11e5829d06ec4b8d1f61.

http://dx.doi.org/10.6084/m9.figshare.1577557.

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
