# Peer review of "Unequal contribution of native South African phylogeographic lineages to the invasion of the African clawed frog, Xenopus laevis, in Europe"

_PeerJ, doi:10.7717/peerj.1659_

## Round 0.1 · original submission · Minor Revisions

The reviewers are supportive of the publication of this manuscript. Please address the issues they have highlighted. Be sure to answer the query about the FST tests raised by the reviewers.

Reviewer 1 ·

Basic reporting

Writing basically fine, though it could do with the fine-toothed comb to rid it of the occasional typo/gramatical lapse.

Experimental design

Generally looks good; see general comments.

Validity of the findings

Generally looks good; see general comments.

Additional comments

Review: “Unequal contribution of native South African phylogeographic lineages to the invasion of the African clawed frog, Xenopus laevis, in Europe" (#2015:10:7102:0:0:REVIEW)

This is a straightforward paper using mitochondrial and nuclear sequence data to infer to geographic origins of invasive populations of the clawed frog Xenopus laevis in Portugal and in France. It builds on a body of data already available for native and invasive species, adding 80 individuals from 32 different localities in France and Portugal, and 14 from five new native South African localities, each represented 2 mtDNA and 3 nuclear gene fragments. The authors used standard phylogenetic and population genetic techniques: direct sequencing, phasing (nuclear sequences); phylogenetic inference under Bayesian and parsimony based methods (mtDNA); median joining networks and comparison of Fst values between populations summarised using principle coordinates analysis (nuclear). The results are discussed in the context of known and inferred events of import and release of specimens. The main conclusion is that French populations came from different native populations in South Africa, i.e. as independent invasions; Portuguese ones, apparently more typically of results for other areas, are derived from a single source. I have mainly minor comments, with one potential criticism of one of the analytical approaches.

1) The aims are somewhat buried in the introduction, and to my taste not framed in a hypothesis testing framework:
„90 This paper assesses DNA sequence diversity in invasive X. laevis populations from France and
91 Portugal in order to identify their potential source areas.“
I would rather put these in a final paragraph, explaining what might be hypothesised and how particular data might be used to test this.

2) A small detail with regard the parsimony analyses, which seem to be pipelined using R in a somewhat complicated fashion (compared to e.g. using PAUP or TNT).
“MP trees were estimated with a
170 neighbour joining tree as starting tree using the Phangorn package (Schliep 2011) in R 3.1.1 (R
171 Core Team, 2014).”
Why? Surely for a parsimony analysis there is no need to find some kind of optimal starting tree using distance?

3) The main (potential) criticism: I don’t understand the Fst comparison: excuse my ignorance as a phylogeneticist, but I would have thought that these comparisons would be appropriate for independent characters, such as SNPs or microsats sampled from across the genome. The application here uses sequence variation in three sequence fragments in each of which the variable sites are linked. Have I misunderstood here? In that case, clarification would be needed. Are these comparisons actually needed, anyway? The main results seems clear from the (appropriately inferred) phylogenetic trees of mtDNA and the median joining networks: the French ones are nested in different SA clades. Perhaps further clarification here too, then, if these results are really necessary, to explain why.

4) The conclusion is actually a summary and not necessary.

5) Fig 1 caption what do the different colours/codes of the dots represent?

Reviewer 2 ·

Basic reporting

This article adheres to most format requirements although the authors should check some details such as the format of citations and whether PeerJ allows citations to unpublished studies.

Experimental design

The experimental design is good.

Validity of the findings

The findings are valid and well justified.

Additional comments

This study uses molecular data to evaluate the history of two introduced populations of the African clawed frog (Xenopus laevis) in Europe, one in Portugal and another in France. Using mitochondrial DNA and two autosomal markers, and comparisons to data from the native range of X. laevis the authors conclude that the population in Portugal was derived from a single source in the southwestern Cape. In contrast, the conclude that the population in France was probably derived from at least two source populations – one in the southwestern Cape and another in the northern / eastern region of South Africa. Most of the data from the native range was from a previous publication but the authors also generated new data from multiple localities in South Africa from the northern / eastern region.

In general I think the methods deployed are appropriate, the results and interpretation are convincing. I strongly support the publication of this paper after the comments below are addressed.

Comments:
The manuscript would be substantially improved by shortening many of the sections and tightening the writing. For example, there are long (and not very interesting) descriptions of the following:

Lines 195-208: parsimony informative and variable positions in the mtDNA alignment.
Lines 217-225, 230-232. polymorphism at autosomal loci.

Much of this is already summarized in Table 1 and there is no need to repeat the specifics in the text. Also, no need to mention the statistics with the outgroup (X. gilli) sequence(s) included. Better to just focus on the XL sequences which are the key ones for this study. If data in the text are missing from the table, this could easily be added.

In the text you could just state whether diversity was higher or lower in each European population than the source populations according to X, Y and Z statistics and then cite the table.

I do not see any advantage to doing a maximum parsimony analysis. The authors did a Bayesian analysis with an appropriately selected model of evolution – this is adequate; the parsimony analysis just detracts from the main message. On Fig. 2 the support values should be “posterior probabilities” not “Bayesian bootstrap” values.

Other comments:
Abstract
It would be nice to have a short statement about the sample size and number of loci used for this analysis to give perspective on the depth of your investigation.

Methods
There is a lot of omission on why certain data where chosen (why those 3 genes?) and where data came from (X. gilli? and others).

It is unclear in all of the Fst analyses what was defined as a population. It is mentioned earlier that there are “seven division in South Africa”, were these the populations? But what then of the invasive populations? Was France all one population or several? Though it may become clear in the Results section, this needs to be clarified in the Methods section. Justification should be provided for why autosomal netowrks were constructed but not other types of analyses (e.g. phylogenetic trees, Structure analyses, etc). A reasonable reason is that there might be evidence of recombination in the autosomal data (?). This could be tested using options in the program DNAsp.

Results: The order of the results section does not match the order of the method section.
If results are discussed by data type (mtDNA and aDNA) as opposed to method (phylogeny, popgen stats, etc), then I would suggest talking about everything to do with that data type all in one section (i.e. don’t talk about mtDNA in one paragraphs, then 2 paragraphs later start talking about mtDNA again, which is currently how the paper is written). As an alternative it might be useful to discuss results by method, especially since the two data types tell similar stories. Also, the median-joining networks are hardly mentioned and only with reference to allelic diversity. The results of that analysis should be explicitly stated “In the median-joining networks allelic diversity clustered…”. Finally, I do not believe that the PCoA was discussed anywhere in the results section, which it should be.

Discussion
Some of the country specific information presented is redundant to the results section. Also, it would be nice to have a small section comparing the X. laevis invasion to that of other species. This paper is very focused on one species and could benefit from a little comparison, broadening the view. Throughout much of the paper, the idea that a species invasion potential is based on its genetic history are not discussed in detail. For example, what are the potential ecological implications of the French population having two different origins? Has this been seen in other species (e.g. invasive populations of anolis lizards in Florida)? What could this mean for the future of the X. laevis invasion?

Line 12: Don’t use an acronym (“nDNA”) without first defining it (and avoid acronyms entirely in the abstract).

line 34/35 Crucial and critical in the same sentence is a bit redundant.
line 37 I think there is a double space between “key” and “to”
line 37 Another acronym with no definition. What is IAS?

Line 43/44 Du Preez et al. 2009 wasn’t studying invasive populations. I would remove the citation.

Line 45 Reducing reproduction of what? Please clarify

Line 48 “indirect negative effects” on what? Please clarify



Lines 49-51 : That the spread of Bd has been linked to invasive species such as XL is an overstatement of the results. I suggest softening this. The Weldon et al. 2004 paper basically only reports detection of chytrid in old specimens. Better to reword that some people have suggested a link, but that it has yet to be established fully (for example see Farrer et al. (2011) Multiple emergences of genetically diverse amphibian-infecting chrtrids include a glocalized hypervirulent recombinant lineage. PNAS). Also, Bd is introduced without an explicit statement that it is an amphibian pathogen (easily clarified by saying “amphibian skin disease chytridiomycosis”). Perhaps citing some papers linking Bd to certain cases of amphibian declines or papers on the negative impact on individuals, etc. For instance: Crawford AJ, et al. (2010) Epidemic disease decimates amphibian abundance, species diversity, and evolutionary history in the highlands of central Panama. PNAS

Line 61/62 Drought and dehydration seem redundant to me.

Lines 68-69, 70-73. Not exactly. Niewoudtville and Kimberley are both on top of the escarpment. Laignsburg and Beaufort West are actually divided by a relatively flat section (not by mountains).

Line 79 Awkward wording. “Given its wide native geographical, ecological range and high…”. Fix wording (“geographical and ecological ranges and high…” might be better”).

Line 89 Does PeerJ allow citation of unpublished manuscripts?

Line 100 Here there could be statements of expectations or hypotheses to outline what the reader should look for in the results.

Line 103 Informal and awkward writing: “whole area known to be invaded by…”. Reword.

Line 106 Native range of what? Please clarify.

Line 108 Shipped where? You are studying multiple locations; Also, personal communication is better than “unpublished manuscript”.

Lines 112 – 116 For what purpose were these localities grouped? Is this for FST? Also please state whether these data were downloaded from Genbank.

Line 126. Please state whether and how animals were euthanized from invasive individuals. Also if museum specimens were prepared, please state where they are deposited.

Line 129 Depending on journal guidelines, please define acrynyms such as “ca” at first use.

Line 138. Missing parentheses

Line 150 Probably should should read something like: “All new sequence data were deposited in Genbank (accession numbers in Online Resource 1)”.

Line 160 Please clarify where did you got the X. gilli sequences.

Line 166 How did you calculate these (Tracer?)?

Line 168 Poorly worded sentence. Posterior probabilities of what (each clade presumably)? Change “remnant” to “remaining”.

Line 178 This is unclear. Fst actually does not require any permutations. The permutations presumably were used to generate confidence intervals.

Line 179 Why is it only “most individuals”? Why were some excluded? This needs clarification.
.
Line 184 Please provide some clarification on what a median-joining network is...most people will not know.

Line 196 It was not mentioned in the methods how parsimony informative sites were quantified.

Lines 194—203 Were there any alignment gaps?

Line 202 Discussion of Fst might be better moved to the end of the paragraph.

Line 210 Suggest changing “They to “These analyses”.

Line 211 – 215 The “data showed” phrasing is fairly informal. I suggest rewording to something like “The phylogeny generated from the 16s data strongly supported...”.

Line 230-1. Please clarify what the units of haplotype diversity are.

Line 232: same for nucleotide diversity

Line 232 A statement about “low and high” needs some level of expectation to be explained (is 0.0016 actually low? Is 0.0059 actually high? What are the bounds that define “low” or “high”). This might be better phrased in a relative way

Line 233 “(phylo)” is unnecessary.

Line 234 Double brackets not necessary.

Lines 240 – 254 The results from mtDNA and autosomal DNA are reported in a confusing way. I suggest restating to say something like “Pairwise nucleotide diversity was XXX, YYY, and ZZZ for cytB, Prmt6 and AR, respectively and haplotype diversity was AAA, BBB, and CCC for …respectively.

Lines 255 – 267 I suggest also these results to the tree, for example, “The two French haplotypes were each closely related to two different South African haplotypes (report something like segregating sites for each), and correspondingly we part of well supported clades (ref tree)”.

Line 296 Rewording needed “and as such, might be…”

Line 297 Again, this idea of knowing the invasion history and how it may affect the spread of individuals would be nicely complemented by an example. How does the genetic history of individuals affect the invasion potential? This is not intuitively obvious.

Line 301 This sentence could be rewritten for clarity (eg. “Genetic diversity of the Portugal population was lower than across the native range of X. laevis in South Africa. However, this invasive population had high to identical sequence similarity with a subset of this native range, namely from the south-western Cape region”).

Line 309 The phrase “identical sequence similarity” is not optimal. Sequences are either “identical” or “similar”, but not both.

Line 321 “Gene diversity”, I’m not sure what you mean by that (haplotype diversity?).

Line 342 This is redundant information and an almost identical sentence to one a paragraph above (lines 313 -- 314).

Line 346 Please reword sentence, currently does not make sense. “the French breeding facility changed from owner”.

Line 349 This is speculation. Cab you list some sources that support this notion or data to support the notion.

Line 361 This sentence either needs substantial reworking or to be removed. It is pointless to compare genetic diversity between different species as so many factors influence diversity such as population size and the rate of evolution. It would be better to point out a comparison of genetic diversity of invasive species with respect to the diversity of their own native ranges (i.e. is comparing the French population to the western cape population different than comparing some other invasive species to its own native range?).

Lines 364—365 This might be better discussed earlier when introducing the idea of studying invasive species

Line 373 “nice study system” this is subjective and, again, informal wording. Suggest removal. Also, please reword “to which extent” (eg. “the extent to which…”).

Line 378 “This showed…” not explicit. Please refer to your analyses.

References:
Depending on PeerJ format, Journal titles should be italicised.

In some citations, journal names are not capitalized (e.g. Molecular ecology, Emerging infectious diseases) and they should be (Molecular Ecology, Emerging Infectious Diseases).

Table 1
Column headers should be centered, not left justified.

Figure 1
The blue of your rivers is inconsistent.

Figure 2 legend There is no such thing as “Bayesian bootstrap” values. Bayesian analyses generate a posterior distribution of trees, from which summary statistics are computed. It has nothing to do with resampling data (i.e. bootstrapping). Please fix this. Also spelling mistake “where” not “were”. Also X. laevis should be labelled on the tree.

Figure 3 legend should mention that these are nuclear genes. The broader implications of this figure are difficult to grasp. A colour legend referring to location (instead of an alternate figure) might help in understanding what any of this means.

---

## Round 0.2 · accepted · Accept

Reviewer 1 has made some additional comments, which you may want to consider. Overall, I am happy that you have addressed the concerns of the reviewers.

Reviewer 1 ·

Basic reporting

No further comments

Experimental design

No further comments

Validity of the findings

No further comments

Additional comments

I reviewed the previous version of this ms., and in looking through again have focussed on those parts that have been modified in revision. Many of the previous suggestions were to better explain analyses and results. Although I think I now better understand what the authors have done and why, the text is in places still rather difficult to follow.

A new section has been added from line 213, starting “The minimum number of recombination events…” that I found particularly tricky to decipher. The first sentence of this paragraph in any case appears to belong in the results. The phrase “intraspecific relationships among the different alleles” seems to confound two different concepts – gene trees and species trees – and I am not convinced that such graphs can be so interpreted. In general a bit of rephrasing in this paragraph would be useful to improve its clarity. I would suggest the following modification; if it is incorrect, I have failed to understand you, but not for want of trying.

“The minimum numbers of recombination events within alleles were estimated in DnaSP v.5 (Hudson & Kaplan, 1985; Librado & Rozas, 2009). Given putative recombination, median-joining networks (MJN) (Bandelt et al. 1999) were constructed in PopART v. 1.7.2 (PopART, 2015) for each nuclear locus separately to illustrate the differences between alleles and their distribution among invasive and native populations (Mardulyn, 2012). Pairwise Fst values were computed based on the allele frequencies to show to what extent alleles are differently sorted among populations (e.g. where the same alleles are found in different populations but with different frequencies).”

I still think that what is now left of the original conclusions section is unnecessary, and it is not a strong conclusion to the paper.

I would agree with the authors that it is better to present the results of two phylogenetic methods to show that the result is robust to the different assumptions. That the results were consistent seems to indicate that there isn’t a problem with the parsimony approach in this case. The neighbourjoining starting tree for the parsimony analyses still seems to me suboptimal though – you’re relying on the branchswapping/island hopping or equivalent algorithm being effective enough to explore all islands of shortest trees from a single starting tree. A better approach in my view would have been to use multiple differing (e.g. random) starting trees.

Reviewer 2 ·

Basic reporting

This is my second review of this article and I am satisfied that the authors have addressed all of my concerns.

Experimental design

This is my second review of this article and I am satisfied that the authors have addressed all of my concerns.

Validity of the findings

This is my second review of this article and I am satisfied that the authors have addressed all of my concerns.